# A Gastroenterologist’s Guide to Care Transitions in Cystic Fibrosis from Pediatrics to Adult Care

**DOI:** 10.3390/ijms242115766

**Published:** 2023-10-30

**Authors:** Dhiren Patel, Michelle Baliss, Pavithra Saikumar, Laith Numan, Jeffrey Teckman, Christine Hachem

**Affiliations:** 1Department of Pediatrics, Division of Pediatric Gastroenterology, Hepatology and Nutrition, Cardinal Glennon Children’s Medical Center, Saint Louis University School of Medicine, St. Louis, MO 63104, USA; pavithra.saikumar@slucare.ssmhealth.com (P.S.); jeffrey.teckman@slucare.ssmhealth.com (J.T.); 2The AHEAD Institute, Saint Louis University School of Medicine, St. Louis, MO 63104, USA; 3Department of Gastroenterology and Hepatology, Saint Louis University School of Medicine, St. Louis, MO 63104, USA; michelle.baliss@slucare.ssmhealth.com (M.B.); laith.numan@slucare.ssmhealth.com (L.N.); christine.hachem@health.slu.edu (C.H.)

**Keywords:** cystic fibrosis, transition of care, pediatrics, gastrointestinal care, pancreatic enzyme, highly effective modulators

## Abstract

Cystic Fibrosis is a chronic disease affecting multiple systems, including the GI tract. Clinical manifestation in patients can start as early as infancy and vary across different age groups. With the advent of new, highly effective modulators, the life expectancy of PwCF has improved significantly. Various GI aspects of CF care, such as nutrition, are linked to an overall improvement in morbidity, lung function and the quality of life of PwCF. The variable clinical presentations and management of GI diseases in pediatrics and adults with CF should be recognized. Therefore, it is necessary to ensure efficient transfer of information between pediatric and adult providers for proper continuity of management and coordination of care at the time of transition. The transition of care is a challenging process for both patients and providers and currently there are no specific tools for GI providers to help ensure a smooth transition. In this review, we aim to highlight the crucial features of GI care at the time of transition and provide a checklist that can assist in ensuring an effective transition and ease the challenges associated with it.

## 1. Introduction

Improved understanding of Cystic Fibrosis (CF) and advances in life-extending treatment over the years have led to considerable changes in the epidemiological profile. While CF was previously considered a lethal pediatric genetic condition, people with CF (PwCF) are now surviving well into adulthood. With increased life expectancy, PwCF face new challenges of living as adults with adult disease manifestations [1]. This improvement in median predicted survival to approximately 50 years of age has introduced new therapeutic challenges and underscored the need for coordinated, uninterrupted, and developmentally appropriate transitions from pediatric to adult-oriented healthcare [2]. Navigating this transition can be highly complex for both patients and providers and carries numerous challenges related to patient readiness, poor insight into this complex disease, psychosocial factors (including mental health issues and adverse social circumstances), changes in disease phenotypes with age, and treatment variations [3,4]. General tools are available to guide patients and providers in constructing a structured transition program (such as “Got Transition” to help guide care coordination improvement and “CF R.I.S.E” as an objective measure of patient readiness); however, no standardized processes currently exist, and current tools are not specific to GI transitions of care [5,6].

CF care requires the involvement of various specialists who play vital roles in transitions of care. The careful collaboration between multidisciplinary CF pediatric and adult teams is therefore imperative to ensure a successful transition, avoid gaps in care, and optimize outcomes. Gastrointestinal (GI) manifestations in particular contribute substantially to extra-pulmonary morbidity in CF and represent a major target for efforts to enhance care transitions [7]. GI manifestations can be directly related to the underlying disease process or a consequence of treatment. Additionally, GI manifestations can impact disease outcomes through their effects on nutrition, pulmonary function, and overall patient wellness. Unfortunately, there is limited awareness of CF disease presentations and management in general adult medicine and an increasing need for pediatric and adult gastroenterologists who focus on the unique GI manifestations of CF. The recognition and management of complex GI pathology in CF requires coordinated expertise given the frequently atypical disease presentations and variable management approaches. The Cystic Fibrosis Foundation (CFF) has developed multiple tools over the years to tackle these issues. These include the Program for Adult Care Excellence (PACE) awards, the Learning and Leadership Collaboratives (LLC), and the Developing Innovative Gastroenterology Specialty Training (DIGEST) Program [4]. Through the DIGEST initiative, the CFF offers training awards to support both pediatric and adult gastroenterologists interested in advancing the care of PwCF. However, there is currently no standardized management and care plan for GI issues in PwCF as they transition into adulthood.

This article aims to highlight the various GI manifestations of CF in the pediatric and adult patient populations and propose a systematic checklist-based approach to improve the transition from pediatric to adult GI care.

### 1.1. Esophagus and Stomach

#### 1.1.1. GERD and Foregut Dysmotility

Gastrointestinal reflux disease (GERD) is a common extrapulmonary manifestation in CF. Several studies have suggested that GERD in CF leads to more severe pulmonary manifestations, though the interplay is complex [8,9,10,11,12]. Multiple underlying mechanical and chemical factors have been implicated in the pathogenesis of GERD in PwCF, including increased transient lower esophageal sphincter (LES) relaxation, increased intra-abdominal pressure from coughing, delayed gastric emptying, decreased basal LES tone, lung hyperinflation effects on the diaphragm, and medication effects (Figure 1) [10,13,14,15,16]. Given the lack of specific guidelines for GERD in CF, the diagnostic and therapeutic strategy for GERD in PwCF does not differ from the general population and includes behavioral modifications, dietary changes, medications, and procedures targeting acid production [17].

GI dysmotility is likely a key factor in the GI manifestations of CF with complex pathophysiology and symptoms that overlap with GERD and dyspepsia. Disruptions in motility are thought to arise from defective epithelial CFTR in the neuromuscular complex acting as a primary insult and leading to altered intestinal myenteric ganglia and abnormal ileal smooth muscle function. Secondary causes of GI dysmotility include decreased bicarbonate and mucoviscidosis from pancreatic insufficiency (PI), fat malabsorption due to poor micelle formation, gut inflammation due to dysbiosis, and hormonal dysregulation [18]. This is further exacerbated by frequent use of anti-cholinergic medications and opiates. Wireless motility capsules have been used to study GI transit profiles in CF and demonstrated significant delay in small bowel transit time without a compensatory increase in whole gut transit time and deficient buffering capacity required to neutralize gastric acid in the proximal small bowel [18,19]. Gastroparesis is a common dysmotility diagnosis in CF that can lead to reduced oral intake, nutritional deficiencies, low body mass index, poor quality of life, and variable response to pancreatic enzyme replacement therapy (PERT). The use of gastric emptying scintigraphy in CF is an area of debate due to inconsistent findings and lack of reproducibility [18]. Specific guidelines for the management of gastroparesis in CF have not been established.

##### Pediatrics

In children with CF (CwCF), GERD is the most common GI manifestation. GERD in children can be exacerbated by the risk factors in Figure 1 and by excessive coughing and use of vest therapy for pulmonary toileting [10]. About two-thirds of children are asymptomatic, while others manifest symptoms of heartburn, abdominal pain and vomiting. Refusal to eat can be a subtle sign of underlying GERD. Higher incidence of aspiration pneumonia is noted in children with symptomatic GERD [20]. It is essential to treat GERD effectively, given the possible association with poor pulmonary outcomes [21]. Unrecognized GERD can limit caloric intake contributing to malnutrition particularly in children, which could be significant for CwCF with higher caloric needs. GERD is a clinical diagnosis, although pH Multichannel Intraluminal Impedance (MII) studies could provide objective evidence. CwCF are on high-dose NSAIDs for anti-inflammatory and protective effects on the lungs, which can adversely cause NSAID-induced gastropathy (gastritis, peptic ulcer disease). Hence, in patients with persistent symptoms despite proton pump inhibitors (PPI) therapy or red flags such as hematemesis, poor growth, esophagogastroduodenoscopy (EGD) should be performed to evaluate for mucosal pathology. Dysmotility in CF affects various portions of GI tract, leading to reflux, dyspepsia, gastroparesis and constipation and can also contribute to refractory reflux symptoms [21]. A systematic review showed a pooled prevalence of gastroparesis of 38% in CwCF, higher than the general pediatric population [22].

Lifestyle modifications such as a healthy diet, avoiding spicy food, limiting carbonated/sugary/caffeinated beverages are emphasized. Behavioral changes, such as avoiding head-down postural drainage in infants, can also help to minimize reflux and aspiration. Commonly used acid reducing medications, in children are histamine receptor 2 antagonists (H2-RA) and PPI. Most children remain on acid suppressive therapy long-term, with attempts to wean if asymptomatic and meeting appropriate growth parameters. PPIs are considered to improve efficacy of PERT and used as an adjuvant in management of PI although its true efficacy in modern PERT use is unknown. Prokinetic agents such as metoclopramide, domperidone, and macrolide antibiotics have low efficacy in treating reflux but are commonly used in treating gastroparesis in children [23]. Azithromycin, a potent anti-inflammatory and pro-kinetic agent, is used to improve gastric motility, although studies have shown it reduces GERD and bile aspiration (specifically after transplantation) in a small subset of CF [24]. There are controversial data regarding efficacy of fundoplication in CwCF [25,26], although it may be considered in medically refractory GERD with comorbidities such as frequent pulmonary exacerbation, recurrent aspiration pneumonia and malnutrition. Apart from reflux esophagitis, other complications such as Barrett’s esophagus and esophageal cancers are rarely seen in the pediatric population.

##### Adults

In adults with CF (AwCF), there is an increased risk of GERD with an estimated prevalence of 55–90% [27]. Like CwCF, regurgitation is a common symptom of GERD in AwCF. On the other hand, heartburn is more commonly reported in adolescents and adults with CF. Abdominal pain or dyspepsia in adults can signify the presence of other gastroduodenal pathology such as gastritis, duodenitis, or peptic ulcer disease, which may warrant further evaluation. Dysphagia could be a sign of esophageal dysmotility or esophageal strictures. Concurrent foregut dysmotility or gastroparesis can manifest with post-prandial early satiety, bloating, nausea, and vomiting and can be difficult to clinically distinguish from GERD. Adults with CF on long-term steroids or immunosuppression following transplant are at an increased risk of infectious esophagitis, which may manifest with odynophagia [28].

As with CwCF, GERD in adults with CF is usually a clinical diagnosis if alarm features (i.e., dysphagia, anemia, weight loss) are absent, and routine management involves dietary and lifestyle modifications and pharmacological acid suppression with PPIs or H2-RAs. Like CwCF, adults with suspected GERD rarely undergo endoscopy for the diagnosis of GERD. Long-term PPI therapy may not always be necessary in AwCF. Thus, during transitions of care, reassessing the indication for PPI use and considering trials of PPI weaning or discontinuation can minimize unnecessary long-term use.

Routine screening for failure to respond to PPI or for the presence of alarm symptoms should be performed at the time of transition. This is especially important considering the substantially increased rate of esophageal adenocarcinoma and Barrett’s esophagus in AwCF compared to the general population with an earlier average age of onset [29]. Individuals on long-term PPI with inadequate symptom control, intolerance to medications, alarm features, or complications such as Barrett’s esophagus or esophageal strictures should be carefully considered for upper endoscopy and for surgical management if clinically appropriate. In patients with dyspepsia or suspected peptic ulcer disease who are at high risk for endoscopy, non-invasive testing for Helicobacter pylori such as stool antigen test or breath test may also be considered. Due to frequent symptom overlap and despite lack of clear guidelines for the work-up and management of dyspepsia and dysmotility in AwCF, further endoscopic or motility testing can be considered in those who fail to respond to PPI. Though data are lacking, a systematic review of the literature noted an increasing frequency of gastroparesis in CF with increasing age [22]. This is thought to be due to the increased risk of diabetes in AwCF. The optimal diagnostic approach for gastroparesis in CF remains an area of debate; however, scintigraphy can be considered. If gastroparesis is confirmed, the use of prokinetic agents such as metoclopramide and macrolide antibiotics can be considered as with CwCF. Decompressive gastrostomy tube and gastrojejunostomy feeding tube placement can be considered in cases refractory to medical therapy.

##### Transition Take-Home Points


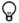
 Reassess the indication for PPI and consider discontinuation trial.


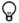
 Screen for alarm features or poor response to therapy and consider upper endoscopy and surgical referral when clinically appropriate due to a higher risk of Barrett’s esophagus and esophageal cancer.

### 1.2. Intestine

#### 1.2.1. Small Intestinal Bacterial Overgrowth

There is a well-established correlation between small intestinal bacterial overgrowth (SIBO) and CF in both children and adults. In both populations, establishing a diagnosis can be challenging. The prevalence of SIBO in individuals with CF has been documented to be up to 56% [15,30,31]. The proposed mechanism of SIBO in CF is attributed to a combination of dysmotility, altered surgical anatomy, frequent antibiotic use, and the accumulation of thick mucus secretions in the small bowel acting as a bacterial anchor. Symptoms in children and adults are similar and can include diarrhea, abdominal pain, flatulence, and weight loss. SIBO in CF may carry nutritional implications related to malabsorption and symptom severity.

##### Pediatrics

SIBO has been reported in almost half of PwCF [31]. Although duodenal aspirate (bacterial count >10^6^ colony forming units/mL and culture) is the gold standard, it is rarely conducted in children. Non-invasive testing, like the hydrogen breath test, is not reliable and difficult to perform in younger children. Diagnosis of SIBO in CwCF is mostly based on clinical symptoms [30,32,33]. In children, symptoms of SIBO can be hard to differentiate from symptoms of malabsorption. If children have prolonged diarrhea, flatulence, poor weight gain despite good caloric intake and optimal PERT, SIBO should be considered [34]. Treatment includes a trial of oral antibiotics. Osmotic laxatives can be beneficial in treating SIBO by preventing stasis and increasing bacterial excretion [12,35]. Prebiotics and probiotics are safe and effective in CwCF [36]. Efficacy of other medications such as N- acetyl cystine and inhaled ipratropium are still under research [30,35].

##### Adults

Adults with CF are at higher risk for recurrent SIBO due to the longer disease duration and recurrent antibiotics exposure, which alters the gut microbiota over time [37]. In AwCF, symptoms include bloating, flatulence, diarrhea, weight loss and malnutrition. Nutritional deficiencies in SIBO include fat-soluble vitamins, iron, vitamin b12. The diagnostic methods are similar to what we have reviewed in children. Empiric treatment of SIBO with rifaximin has also been proven to be effective, well tolerated and currently is the gold standard treatment in both adults and pediatrics, although lack of reliable health insurance coverage can lead to treatment delay. [38,39]. During care transitions, PwCF with nutritional deficiencies or symptoms of SIBO should be considered for diagnostic evaluation or empiric therapy.

#### 1.2.2. Intestinal Obstruction Syndromes: Meconium Ileus, DIOS, Constipation

Intestinal obstructions syndromes are interrelated and include meconium ileus at birth, distal intestinal obstruction syndrome (DIOS), and constipation. They occur mainly due to increased or decreased transit time, PI, and the high mucus viscosity in the intestinal tract which leads to intestinal obstruction [40,41]. Variable treatment options are utilized but none have good supporting evidence. More randomized controlled trials are needed.

##### Pediatrics

Around 15% of infants with CF are diagnosed with meconium ileus (MI), which commonly presents in the neonatal period when inspissated meconium causes obstruction. [42]. Simple MI is treated with hyperosmolar enema (gastrografin, N- acetylcysteine enema), while complicated MI (with perforation, peritonitis, volvulus) needs surgical intervention. During transitions of care, it is important to highlight if the patient has a history of MI and if it required surgical intervention due to potential complications and increased risk for DIOS.

DIOS is a unique condition in CF that presents as acute, complete or partial fecal obstruction, more commonly in the ileocecal region. Patients with severe CFTR mutation, history of meconium ileus, PI, history of prior DIOS and CFRD are at higher risk for DIOS [43,44]. Adequate management of chronic constipation and optimization of PERT has been noted to prevent DIOS [45,46,47]. Dietary avoidance of bulky high fiber foods should be advised. Medical management includes IV fluids, laxatives (polyethylene glycol) and gastrografin enemas. Surgery is recommended for failure of conservative management or complications like perforation, peritonitis, or sepsis and should be avoided if possible [41].

About 50% of CwCF have chronic constipation [44], due to risk factors such as altered intraluminal fluid-ion composition, dysmotility, and dysbiosis. Constipation has a gradual symptom onset compared to DIOS, which presents acutely. Although there is lack of evidence, exercise and adequate water intake are emphasized. Daily laxative regimens consisting of osmotic laxatives (preferably PEG) are advised. Stimulant laxatives like senna and bisacodyl are considered second line. Failure to maintain a daily laxative regimen and early signs of DIOS should prompt an increased dose of osmotic/stimulant laxatives. Enemas may be utilized to avoid complications and the need for hospitalization.

##### Adults

While meconium ileus is an exclusively pediatric condition in CF, DIOS, formally known as a meconium ileus equivalent, is more commonly seen in AwCF with a peak incidence between 20–25 years of age [47]. The definition was specified by the European Society for Pediatric Gastroenterology, Hepatology, and Nutrition (ESPGHAN) to make a clear distinction between DIOS and constipation [43]. DIOS has varying presentations ranging from incomplete obstruction to acute complete obstruction. Treatment is mainly conservative with fluids, nasogastric decompression, and enemas [45]. Surgical intervention may be warranted with failure to respond to conservative management or with signs of perforation or ischemia [48].

As with pediatrics, constipation is one of the more common GI symptoms in CF, and prevalence increases with age. Compared to DIOS, constipation has a more gradual and subtle presentation but it can be challenging to differentiate from DIOS [49]. Osmotic laxatives are the initial treatment for AwCF with constipation. Stimulant laxatives can be used as an additional treatment [50]. If the initial treatment fails, other options include Lubiprostone (chloride channel activator) and Linaclotide (guanylate cyclase-C agonist) [49,51,52]. Limited data exist on the best option, as few have been studied in PwCF.

During transitions of care, providers should identify individuals at risk of DIOS including those with prior surgical intervention for meconium ileus or PI. For individuals with a history of constipation, obtaining a thorough history of previously tolerated and unsuccessful treatments can guide future care. Optimizing PERT dosing, adjusting bowel regimens and adding chloride channel activators or other bowel regimens when indicated can help prevent DIOS and relieve constipation.

##### Transition Take-Home Points


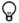
 Identify patients with nutritional deficiencies with symptoms of SIBO and consider for diagnostic evaluation or empiric therapy.


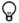
 Identify patients with a history of meconium ileus requiring surgical intervention during childhood or PI who are at higher risk of DIOS and constipation.


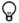
 Obtain a thorough history of previously tolerated and unsuccessful treatments for constipation to guide future care which can include optimizing PERT dosing, adjusting bowel regimens, and consideration of combination therapies.

## 2. Pancreas

### 2.1. Pancreatic Insufficiency

The CFTR gene plays a significant role in pancreatic ductal secretion. Pancreatic disease in CF varies in phenotype and severity depending on the type of CFTR mutation and the degree of loss of function. Those who are homozygous with severe mutation variants (classes 1, 2, 3) develop PI as early as birth in comparison to milder mutation variants (class 4, 5) or heterozygous individuals. Individuals who retain some pancreatic functions are at a higher risk of acute pancreatitis than those with PI [53,54].

#### 2.1.1. Pediatrics

Pancreatic injury from ductal obstruction starts in utero and most individuals are diagnosed with PI by 1 year of age [55]. Pancreatic exocrine deficiency manifests with poor weight gain, diarrhea, failure to thrive, and fat-soluble vitamin deficiency. As children develop PI, it is very important to ensure nutritional optimization with a high calorie diet and early PERT.

CFF guidelines recommend starting PERT in infants who have fecal elastase <200 and or signs of malabsorption [56]. Dosing of PERT is high in infants (2000–5000 mg lipase per 4 oz of formula/breastmilk) which is then regulated to a weight-based dosing after 2 years of age. Dosing is titrated based on malabsorption (clinical symptoms and growth) and amount of fat intake. Optimization of lipase dosing is essential to prevent fibrosing colonopathy, which is a rare long-term complication of excessive lipase dosing [57].

CF related diabetes (CFRD) rarely manifests in younger children. Roughly 20% of PwCF develop CFRD during adolescent age. Annual screening with OGTT (oral glucose tolerance test) is recommended starting at age 10.

#### 2.1.2. Adults

Since 85–90% of PwCF carry CFTR mutations resulting in impaired or complete loss of CFTR function, the majority of adults with CF carry a diagnosis of PI from early childhood [58]. PI in adults presents similarly to the pediatric population with frequent malodorous, oily, floating stools and usually necessitates lifelong PERT (dosed in adults either based on grams of ingested fat or on weight and titrated up as needed for malabsorption). It is important to note that patients born with pancreatic sufficiency (PS) may experience progressive loss of acinar cells and can eventually develop PI during adulthood [53]. It is recommended by the CFF that individuals with PS should be screened for PI annually. During transitions of care, routine screening for PI should be performed using fecal elastase, especially if not previously documented. In addition, it is important to assess patient understanding regarding the correct use of PERT and to screen for factors affecting adherence.

Contrary to exocrine function, pancreatic endocrine function in PwCF tends to be preserved during childhood but eventually can become impaired due to progressive islet cell destruction [59]. This leads to CFRD in 50% of adults and 20% of adolescents with an average age of onset of 18–21 years [58]. Given predominantly adolescent and adult onset, it is prudent to ensure screening for CFRD at time of care transition, as early diagnosis and treatment are essential in morbidity reduction.

### 2.2. Pancreatitis

Acute pancreatitis is uncommon in PwCF due to the widespread destruction of the exocrine pancreas in utero as early as the seventeenth week of gestation, especially in homozygous variants of the CFTR mutation [58,60]. This leaves little to no residual pancreatic enzyme capable of producing inflammation in PI. Acute pancreatitis is typically seen in people with heterozygous CFTR mutations or milder variants with preserved pancreatic function.

#### 2.2.1. Pediatrics

Pancreatitis can be one of the initial presentations of CF, especially in the milder variants [61]. In children with >1 episode of acute idiopathic pancreatitis, genetic testing for mutations including CFTR is recommended. The incidence of acute pancreatitis in pancreatic sufficient (PS) children is 10% in comparison to 0.5% in pancreatic insufficient children [62]. Even in those with PS, recurrent acute pancreatitis can progress to PI and CFRD [63]. There are case reports of acute pancreatitis in children with homozygous mutations during or after treatment with lumacaftor/ivacaftor [64]. Modulator therapy may improve pancreatic acinar mass and function, which is supported by elevated fecal elastase [65,66]. During transitions of care, it is important to communicate duration of modulator therapy, whether the patient is PS or PI, and an assessment of prior episodes of acute pancreatitis.

#### 2.2.2. Adults

As with children, acute pancreatitis is almost exclusively seen in PS with acinar tissue reserve and typically presents in teenage years or adulthood. Among adults with PS, it is estimated that 20% will experience pancreatitis, 18% of which presents as a single episode of acute pancreatitis, 60% with acute recurrent pancreatitis, and 22% progress to chronic pancreatitis [67]. Despite the low prevalence of acute pancreatitis in CF, it is important to identify individuals with PS during care transitions in whom a high suspicion for acute pancreatitis should be maintained when presenting with abdominal pain.

#### 2.2.3. Transition Take-Home Points


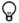
 Identify PwCF and pancreatic sufficiency by checking fecal elastase.


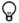
 Acute pancreatitis should be considered as a potential cause of abdominal pain in individuals with pancreatic sufficiency.


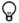
 Screen for CFRD at time of the transition with oral glucose tolerance test, as the risk increases with age due to progressive pancreatic islet cell destruction.

## 3. Hepatobiliary

CF liver disease (CFLD) is a term used to describe the wide range of pathologies affecting the hepatobiliary system including neonatal cholestasis, cholelithiasis, sclerosing cholangitis, micro-gallbladder, gallbladder dyskinesia, hepatic steatosis, hepatitis, focal biliary cirrhosis, multi-lobular biliary cirrhosis, and cirrhotic and non-cirrhotic portal hypertension [68,69,70]. CFLD is the third most common cause of mortality among PwCF [42]. Potential etiologies include increased bile viscosity, low innate immunity to endotoxins leading to toxic bile salt accumulation and gut dysbiosis [70].

### 3.1. Pediatrics

The prevalence of CFLD is high among CwCF, ranging from 5% to 68% based on the disease manifestation [71]. Focal biliary cirrhosis progressing to multi-lobular cirrhosis is seen in 5–10% of CwCF in the first decade [72]. Risks for CFLD is higher in males and those with PI, CFRD, history of meconium ileus, and genetic factors [73,74]. Diagnostic criteria includes two of the following: clinical evidence of hepatomegaly or splenomegaly, persistently elevated liver enzymes (>12 months on three different occasions), abnormal liver ultrasound findings, and abnormal histopathology [68]. Guidelines recommend annual liver enzymes to assess liver function and further investigations for any abnormality. Ultrasound and transient elastography can be a non-invasive modality to assess for fibrosis, but its reliability in identifying early fibrosis, especially in younger children is questionable. Treatment comprises supportive therapy and monitoring/treating complications of cirrhosis and portal hypertension. Surveillance EGD are not routinely performed for asymptomatic children even with varices unless they present with bleeding. A study found insufficient efficacy of ursodeoxycholic acid (UDCA) use in CFLD, although it is often clinically used for cholestasis from CFLD [75].

Children with CFLD are at higher risk for malnutrition and nutritional deficiencies. Including dietary medium chain triglycerides and poly-unsaturated fatty acids can provide 130–150% of required daily calories [55]. Fat-soluble vitamin supplementation and regular monitoring to avoid toxicity is essential in CFLD.

### 3.2. Adults

CFLD is well defined in pediatrics but less clearly described in adults, accounting for 5% of CF mortality [76]. However, with the improved life expectancy in CF, CFLD is becoming more prevalent in adults and encompasses a similar spectrum of liver diseases as reviewed in CwCF [69,76,77,78]. Liver diseases are mainly asymptomatic until they are in advanced stages; therefore, early detection is key. Annual screening for CFLD with history, physical examination, laboratory testing, and non-invasive imaging (ultrasound or fibroscan) is recommended [76].

For established CFLD, management involves avoidance of hepatotoxic substances/medications, optimizing nutrition, and managing complications of portal hypertension or liver failure [68]. Key management items related to severe liver disease include limiting contact sports (due to splenomegaly), NSAID avoidance in portal hypertension, awareness of cholestatic pruritis, and awareness of sleep and attention disturbances of early hepatic encephalopathy. Liver transplantation is the ultimate treatment for advanced liver disease in CF; however, the timing is a challenge due to progressive pulmonary disease and malnutrition [68]. Early referral is recommended to minimize complications.

### 3.3. Transition Take-Home Points


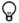
 Identify PwCF with previous liver disease or injury and obtain an initial assessment of their liver function and any signs of chronic liver disease.


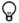
 Screen for and counsel regarding modifiable risk factors for hepatotoxicity such as alcohol use or recreational drug use.

## 4. Nutritional Failure

### 4.1. Pediatrics

Being underweight and failure to thrive are common manifestations in younger kids with CF [54]. Multiple studies have shown a correlation between malnutrition and poor lung function [79]. Conversely, with use of highly effective modulator therapy, obesity and disordered eating are on the rise among older kids and adolescents [80].

Due to malabsorption and increased caloric needs, children need ~130–150% of the recommended daily allowance (RDA) for positive energy balance to promote growth. Daily caloric requirements are calculated based on activity levels in older kids. Most of the calories should come from carbohydrates (50%) and the rest from protein (35–40%) and fat (15–20%). Poor absorption is an important cause for malnutrition; hence reviewing the dose of PERT is essential.

Micronutrient deficiencies include fat-soluble vitamins, zinc and essential fatty acids should be measured to prevent complications and subsequently monitored to avoid hypervitaminosis (Table 1) [81].

Due to the immense focus on nutrition and PERT, obese and overweight individuals have quadrupled in the last decade [82]. CFTR modulators are reported to increase weight in CF by improving appetite and reducing malabsorption. These medications also increase HDL, LDL and cholesterol as well as improve insulin secretion [83,84]. It is important to monitor for signs of overnutrition and perform screening labs such as lipid panels annually.

### 4.2. Adults

Malnutrition has previously been one of the most common manifestations of AwCF, and survival is linked to nutritional status [85]. Corey et al. [85] compared the median survival of two large CF clinics with different dietary approaches. Patients with a high-calorie, high-fat diet had better nutritional status and improved median survival. This led to the recommendation of high-calorie, high-fat and dietary supplements to help maintain nutritional status in CF [86]. However, most of the clinical studies were focused on children. There has been limited data on nutrition in AwCF. A retrospective analysis of 81 AwCF revealed that malnutrition is still a prevalent issue and that a more aggressive nutritional approach is required [87]. Given the improved survival with better nutrition and highly effective modulator therapy, a complete review of current and previous dietary habits and supplements is warranted. The initial transition visit should include a comprehensive nutritional assessment and counseling to intervene early if there are any signs of malnutrition. Supplemental enteral feeding when non-invasive approaches fail to improve nutritional status may be required. Some studies looked at the benefits of enteral feeding in CF, and these studies showed evidence of weight gain, improved body composition and improved pulmonary function in one of the studies [88,89,90].

On the other hand, a high-calorie and high-fat diet may increase the risks of obesity, cardiovascular complications, CFRD, and fatty liver disease. Data is limited on the detriments of obesity in PwCF. A study showed better pulmonary function in obese or overweight PwCF when compared to normal or underweight individuals and suggested a cutoff BMI < 30 kg/m^2^ [91]. Providers should assess for obesity at the time of transition to minimize future complications.

### 4.3. Transition Take-Home Points


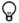
 Obtain a complete review of previous dietary habits and supplements.


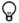
 Evaluate for signs of malnutrition, obesity/overweight and disordered eating during the initial transition visit.

## 5. GI Cancers

With the improved life expectancy of PwCF, there is an increased prevalence of GI malignancies, including esophageal, colorectal (CRC), small bowel, pancreatic, and hepatobiliary [92]. This risk is significantly higher than in the general population, especially with severe CFTR mutations and following lung transplantation [93]. The most common GI malignancy in AwCF is CRC; CF may be classified as a hereditary colon cancer syndrome [94]. It is estimated that AwCF are at a 5–10 times higher risk of CRC than the general population and a 25–30 times higher risk following organ transplantation [95]. The median onset of CRC in PwCF is estimated to be 20–30 years younger than the general population. The increased prevalence and rapid progression of GI malignancies is thought to be precipitated by the altered expression of the CFTR gene and is one of the most devastating aspects of adult CF GI care [94]. Guidelines for screening for GI malignancies in CF (other than CRC screening) remain limited, and there is certainly a need for standardized screening strategies based on specific risk factors.

The current consensus on CRC screening in PwCF is that the decision should be individualized based on comorbidities and patient-provider discussion. Colonoscopy is currently the preferred screening modality, but non-invasive methods are currently being investigated. Screening should begin at age 40 and be repeated at 5-year intervals or sooner, depending on individual findings. Individuals who have undergone organ transplantation and are 30 years of age or older should begin CRC cancer screening within 2 years of the transplant.

Despite lacking guidelines on screening for other GI malignancies, the increased risk of malignancy should be recognized as children transition into adulthood, and careful investigation of any alarm or unexplained GI symptoms should be considered. Individualized screening strategies depend on the burden of other comorbidities, transplant status, and the severity of pulmonary manifestations.

### Transition Take-Home Points


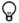
 Carefully investigate for any alarm or unexplained GI symptoms during the transition of care. Individualized GI cancer screening strategies depending on the burden of other comorbidities, transplant status, and severity of pulmonary manifestations should be discussed during care transitions.

## 6. Conclusions

CF is a chronic disease, manifesting clinical features from infancy through adulthood. Ideally, PwCF are taken care of by a multidisciplinary team. The transition of care from pediatric to adult providers can be a challenging path for both patients and providers. With variations in clinical manifestations and treatment between pediatric and adult providers, it is important to identify processes for a smooth transition. We have created a system-based checklist (Figure 2 and Table 2) that provides a standardized process for GI transitions of care. As this is a novel effort, new developments and feedback received from PwCF and providers should be incorporated regularly to continue to optimize gastroenterology transitions of care in CF. This is especially important as the introduction of modulator therapy for CF has created a changing landscape, and gastroenterology manifestations of CF are evolving.

## Figures and Tables

**Figure 1 ijms-24-15766-f001:**
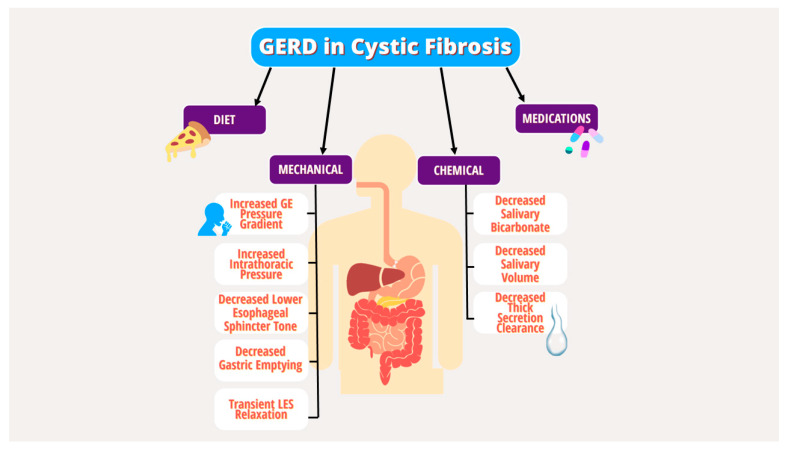
Mechanical and chemical factors in the pathogenesis of GERD in PwCF.

**Figure 2 ijms-24-15766-f002:**
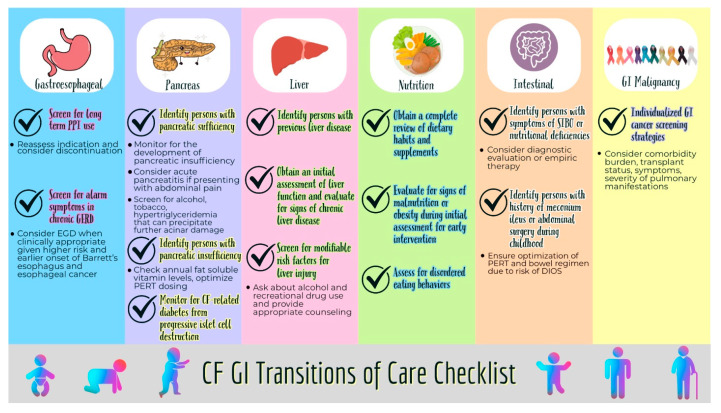
Transition of Care Checklist.

**Table 1 ijms-24-15766-t001:** Micronutrient deficiency in PwCF.

Vitamin A	Increased Infection susceptibility Growth Failure Corneal Opacities Xerophthalmia Bitot Spots Night Blindness
Vitamin D	Bone Pain Motor Delays and Poor Growth Delayed Fontanelle Closure Craniotabes, Frontal Bossing Widening of Ankles and Wrists Bowlegs or Knock Knees Widening of Growth Plate
Vitamin E	Decreased Reflexes and Sensation Ataxia Myopathy Retinopathy Hemolytic Anemia
Vitamin K	Easy Bruising Mucosal Bleeding Splinter Hemorrhages GI and GU Bleeding
Essential Fatty Acids	Scaly Dermatitis Alopecia Thrombocytopenia Poor Growth Poor Cognitive Function Visual Impairment
Zinc	Depressed Immunity Impaired Taste and Smell Acrodermatitis

**Table 2 ijms-24-15766-t002:** Points to remember during transition of care.

Esophagus	Screen for long-term use of PPI, reassess indication, and consider discontinuation trial to reduce unnecessary long-term use. Screen for alarm symptoms in AwCF with long-standing GERD and consider upper endoscopy when clinically appropriate given the higher risk and earlier onset of Barrett’s esophagus and esophageal cancer.
Pancreas	Identify PwCF with pancreatic sufficiency. Monitor for the development of pancreatic insufficiency from progressive loss of pancreatic function. Consider the possibility of acute pancreatitis in PwCF with pancreatic sufficiency presenting with abdominal pain. Screen for hypertriglyceridemia, tobacco use, and alcohol use that can precipitate further acinar damage and result in pancreatic insufficiency. Identify PwCF with pancreatic insufficiency. PwCF with pancreatic insufficiency should have fat-soluble vitamin levels checked annually. Assess understanding of the correct use of PERT and optimize dosing. Monitor for signs of CF-related diabetes from progressive pancreatic islet cell destruction especially in AwCF.
Liver	Identify PwCF with previous liver disease or injury. Obtain an initial assessment of liver function and any signs of chronic liver disease. Screen adolescents and adults for modifiable risk factors for hepatotoxicity such as heavy alcohol use and recreational drug use and provide appropriate counseling.
Nutrition	Obtain a complete review of previous dietary habits and supplements. The initial visit should include a comprehensive nutritional assessment and counseling to intervene early if there are any signs of malnutrition. Assess disordered eating behaviors.
Intestines	PwCF with nutritional deficiencies or symptoms of SIBO should be identified and considered for diagnostic evaluation or empiric therapy for SIBO. Obtain a thorough diagnostic and treatment history for PwCF previously diagnosed with SIBO to help adult providers plan future care. Identify PwCF with a history of meconium ileus requiring surgical intervention during childhood or pancreatic insufficiency who are at higher risk of DIOS and ensure optimization of PERT and bowel regimen.
GI malignancy	Individualized GI cancer screening strategies depending on the burden of other comorbidities, transplant status, and severity of pulmonary manifestations should be discussed.

## Data Availability

This manuscript is a review article with no data collection or analysis to report. All authors confirm that the manuscript is written based on existing literature and expert opinions from the authors of this article.

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
