# Peer review of "A Gastroenterologist’s Guide to Care Transitions in Cystic Fibrosis from Pediatrics to Adult Care"

_ijms, 2023, doi:10.3390/ijms242115766_

Round 1

Reviewer 1 Report

Comments and Suggestions for Authors

In this article, the repercussions of Cystic Fibrosis (CF) on the gastrointestinal (GI) tract are explored extensively, both in pediatric and adult populations. Due to the emergence of potent therapies and modulators, the lifespan of those with CF has notably extended, giving rise to a pressing necessity for adept diagnosis and management of age-related complications in adults, an area that is currently underserved in healthcare.

The authors present an in-depth and articulate review, aiming to bridge the existing knowledge gap regarding the handling of CF-induced GI complications in adult medicine. This calls for a concerted and specialized approach to address the intricate GI pathology that is linked with CF. The review emphasizes the criticality of recognizing the diverse clinical presentations and treatment strategies between pediatric and adult CF populations, highlighting the pivotal role that GI care, inclusive of adequate nutrition, plays in enhancing the well-being, lung functionality, and overall quality of life of these individuals.

Moreover, the article underscores the urgent need for improved collaboration and coordination between pediatric and adult healthcare teams, especially during the critical period of transition of care. Presently, this process is struggling with difficulties, largely owing to the absence of specific instruments that would ensure a seamless transition.

Delving deeply into the major GI manifestations of CF including, but not limited to, GERD, liver disease, DIOS, MI, PI/PS and bacterial overgrowth, the authors offer a meticulous prognosis for both pediatric and adult groups, shedding light on the challenges encountered during transition. Furthermore, the review aims to bring to the fore crucial elements of GI care during this phase, introducing a structured checklist to facilitate a more streamlined and less daunting transition process.

Overall, the review is of very high quality, well written and organized and will add a significant contribution to the field. I highly recommend publication of the article.

Author Response

Thank you for taking your time and reviewing our manuscript. We really appreciate your comments. 

Reviewer 2 Report

Comments and Suggestions for Authors

With limited literature on the transition of gastrointestinal (GI) care in people with CF, this review systematically focuses on the challenges involved in this process. Overall, the review is detailed and written well. My major criticism is that the review elaborates on the GI transition's medical aspects but fails to address the other issues involved. A short paragraph discussing the barriers for transition of care (patient's developmental lag, poor insight of the disease, adverse social circumstances, mental health disorders such as anxiety/depression) can be included. Transition of care has been relatively well studied in other disorders such as IBD, asthma and authors can quote some studies including these details.

Also, the authors need to emphasize that in CF, the transition of GI care often occurs in conjunction with the transition of other specialties such as pulmonary, nutrition, respiratory therapy, endocrinology, etc., and this complexity can be highlighted.

Minor comments:

Page 5 – SIBO (Adults) – I would recommend removing albumin. In CF, it is a negative acute phase reactant and should not be used as a nutritional biomarker.

Author Response

Thank you for taking your time to review the manuscript. 

Your suggestions and comments are greatly appreciated and changes have been incorporated into the article. 

Reviewer 3 Report

Comments and Suggestions for Authors

In this review Patel et al. sumarise the GI issues associated with CF and provide a check list to assist effective transition between Peadiatric to adult care of patient with CF. This review systematically discusses 1- Oesophagus and Stomach issues: GORD and upper GI dysmotility 2- midgut issues: SIBO and DIOS. 3- Pancreatic issues: Insufficiency and inflammation 4- Hepatobiliary issues 5- Nutritional issues and lastly 6- GI Cancers.

This is an interesting and well written review.
Since NSAIDs are used for inflammation and pain management of CF, a more in-depth discussion of the connection between NSAIDs and GORD should be present in the first chapter: noting that the associated reduction of prostaglandins takes off one of the inhibitory mechanisms on acid production in parietal cells.

The structure is clear for each chapter; however the font of the headings is not consistent [some in bold italics or lacking numeration [for instance 1.2.2intestinal Obstruction syndromes...].  This must be revised for clarity. 

Figure 1:  is clear and well designed.

Figure2: the pagination of the table needs attention. Centered pagination is not the best choice in the table. Also: since it is a table, it should be labelled as Table 1.
The second part of figure 2 is a useful figure, however the choice of colors and font are not the best and makes reading some writing challenging.

Table2: The micronutrient deficiencies table is lacking title and legend. The table would be better presented lengthwise and lighter background would facilitate reading. Here again the font colors are not optimal.

In summary this is a good review that should be published after careful review of the pagination and illustration, as well as some details regarding the interaction of GORD and NASIDs for CF patients.

Author Response

Thank you for taking the time to review the manuscript. 

Your suggestions and comments are valuable and changes have been incorporated in the article. You made a great point about NSAID use and related gastropathy, we have included that in our manuscript. All suggestions regarding formatting figures/tables are very useful and have helped make this article reader-friendly. 
